# Assessing the impact of human trampling on vegetation: a systematic review and meta-analysis of experimental evidence

Oliver L. Pescott[1] and Gavin B. Stewart[2]

[1] Centre for Ecology & Hydrology, Wallingford, Oxfordshire, UK
[2] School of Agriculture, Food and Rural Development, Newcastle University, Newcastle upon Tyne, UK

## ABSTRACT

Vegetation trampling resulting from recreation can adversely impact natural habitats, leading to the loss of vegetation and the degradation of plant communities. A considerable primary literature exists on this topic, therefore it is important to assess whether this accumulated evidence can be used to reach general conclusions concerning vegetation vulnerability to inform conservation management decisions. Experimental trampling studies on a global scale were retrieved using a systematic review methodology and synthesised using random effects meta-analysis. The relationships between vegetation recovery and each of initial vegetation resistance, trampling intensity, time for recovery, Raunkiaer life-form (perennating bud position), and habitat were tested using random effects multiple meta-regressions and subgroup analyses. The systematic search yielded 304 studies; of these, nine reported relevant randomized controlled experiments, providing 188 vegetation recovery effect sizes for analysis. The synthesis indicated there was significant heterogeneity in the impact of trampling on vegetation recovery. This was related to resistance and recovery time, and the interactions of these variables with Raunkiaer life-form, but was not strongly dependent on the intensity of the trampling experienced. The available evidence suggests that vegetation dominated by hemicryptophytes and geophytes recovers from trampling to a greater extent than vegetation dominated by other life-forms. Variation in effect within the chamaephyte, hemicryptophyte and geophyte life-form sub-groups was also explained by the initial resistance of vegetation to trampling, but not by trampling intensity. Intrinsic properties of plant communities appear to be the most important factors determining the response of vegetation to trampling disturbance. Specifically, the dominant Raunkiaer life-form of a plant community accounts for more variation in the resilience of communities to trampling than the intensity of the trampling experienced, suggesting that simple assessments based on this trait could guide decisions concerning sustainable access to natural areas. Methodological and reporting limitations must be overcome before more disparate types of evidence can be synthesised; this would enable more reliable extrapolation to non-study situations, and a more comprehensive understanding of how assessments of intrinsic plant traits can be used to underpin conservation management decisions concerning access.

Corresponding author
Oliver L. Pescott,
oliver.pescott@ceh.ac.uk

## INTRODUCTION

Recreational pressure can cause many problems for managers of nature reserves, countryside and wilderness (*Leung & Marion, 2000*). Effective management is essential if the recreational usage of natural habitats is to be balanced with the retention of the nature conservation value of a site. In this context, biodiversity managers and researchers in the UK have identified the need for increased knowledge about the impact of recreational activities on biodiversity as one of the 100 most policy-relevant ecological questions (*Sutherland et al., 2006*); recent differences of opinion regarding open access policies on England's National Nature Reserves underscores the need for robust evidence in this area (*Marren, 2013*). Human trampling, and the response of vegetation to this disturbance, have been a main focus of research on sustainable use of natural habitats for recreation, and have been investigated in many different habitat types around the world. Studies investigating vegetation responses to trampling impacts have utilised various methodologies, including descriptive surveys, site comparisons, before-after control-impact (BACI) designs, and experimental approaches (*Sun & Walsh, 1998*). A standard randomised, controlled experimental design has been described by *Cole & Bayfield (1993)*, and is often used by workers in this field. Standardised procedures should allow for greater comparability between studies, especially where scale-dependent variables affect the measurement of impacts significantly (*Taylor, Reader & Larson, 1993*). Work on the impact of trampling on vegetation has been reviewed on several occasions (*Liddle, 1975a*; *Kuss, 1986*; *Yorks et al., 1997*); however, for a transparent and comprehensive synthesis of the available evidence, a systematic methodology should be employed for the retrieval, critical appraisal and pooling of studies (*Pullin & Knight, 2003*; *Sutherland et al., 2004*; *Pullin & Stewart, 2006*; *Stewart, 2010*).

The relevance of trampling studies for conservation managers and practitioners depends on the nature of the managed site, the plant communities contained within, and the type of access in use or being considered (*Burden & Randerson, 1972*; *Cole, 1987*). Some sites may be essentially open access, whilst others may guide or restrict users to paths or delimited areas. Workers studying trampling have divided community responses to trampling into various categories and series with the intention of producing indicators or indices representing the responses of plant communities (*Leung & Marion, 2000*). Resistance, the intrinsic capacity of vegetation to withstand the direct effect of trampling (*Liddle, 1975b*), and resilience, the intrinsic capacity of vegetation to recover from trampling (*Kuss & Hall, 1991*), are most often used as indicators of impact. They allow ecological data to be distilled into categories with biological relevance and conceptually straightforward links to management practice.

Studies on trampling have examined the impacts on physiological & morphological vegetation characteristics (*Kuss & Graefe, 1985*), soil fauna (*Chappell et al., 1971*)

and a range of edaphic variables (e.g., *Andersen, 1995*; *Ros et al., 2004*). However, the response reported most frequently is vegetation cover, which can be used to quantify the vulnerability of vegetation types using measures of resistance and resilience (*Cole & Bayfield, 1993*). Primary studies often present data on vegetation cover as 'relative vegetation cover' (RVC); this is the cover on a trampled plot relative to its initial cover, adjusted for changes in cover on control plots during an experiment (Supplemental Information 1; *Cole & Bayfield, 1993*).

The responses of vegetation to trampling have been reported to be affected by trampling intensity (number of human trampling passes; e.g., *Cole, 1987*; *Cole, 1995a*), frequency (trampling passes per time period; *Cole & Monz, 2002*), distribution (whether trampling passes are dispersed or clumped for a particular trampling frequency; *Gallet, Lemauviel & Rozé, 2004*), season (*Gallet & Rozé, 2002*), weather (*Gallet & Roze, 2001*), habitat (*Liddle, 1975b*), species (*Gallet, Lemauviel & Rozé, 2004*), Raunkiaer life-form (i.e., perennating bud position) and growth-form (*Cole, 1995b*), and soil type (*Talbot, Turton & Graham, 2003*). Here, we consider variation in trampling intensity, vegetation resistance, recovery time, Raunkiaer life-form of the community dominant and broad habitat type as potential reasons for heterogeneity in experimental results across primary studies.

To our knowledge, no attempt has been made at a formal systematic evaluation of the effect of trampling on vegetation, or at a meta-analytical synthesis of available data. Therefore, the aims of this study were to systematically assess and review the evidence for the effects of human trampling on plant communities, to synthesise experimental data via an appropriate meta-analytical technique, and to investigate variables associated with significant variation in study outcomes. Clearly, robust synthesis relies on robust data, and much relevant data on the effects of human trampling on plants may not be suitable for quantitative synthesis (*Yorks et al., 1997*). The nature of the available data constrained the specific questions that we were able to address to: (1) Does resilience (i.e., vegetation recovery) differ with respect to trampling intensity, initial resistance of the vegetation, recovery time, Raunkiaer life-form of the community dominant, or habitat? And, (2), how is resilience affected by these covariates and their interactions? We also present tables and lists of relevant primary studies to promote the future synthesis of the considerable amount of observational and mensurative ecological work performed in this area.

Practical implications and guidance may be able to be derived where the relationships between resilience and covariates (e.g., trampling intensity or time allowed for recovery) can be manipulated by management decisions. We also comment on the limitations imposed on this systematic review by the original papers, and assess the utility of experimental trampling studies for improving the scientific basis of the management of human trampling impacts in areas of conservation importance. Given the large number of studies that have now investigated the impact of human trampling on plant communities (*Yorks et al., 1997*; Supplemental Information 3 and 4), it is important to assess whether the accumulated data available in the primary scientific literature can be effectively mined for ecological patterns providing reliable across-study evidence that can be used to support conservation management decisions (*Sutherland et al., 2004*; *Pullin & Stewart, 2006*).

## METHODS

### Identification of relevant studies

Our methodology follows the approach of *Pullin & Stewart (2006)*. Relevant studies were identified via systematic searches of electronic databases, including: JSTOR, ISI Web of Knowledge, ScienceDirect, Google Scholar, DOAJ, Copac, Scirus and Agricola. English language search terms reflecting elements of the review question were used in mining the databases. The references of relevant articles were hand-searched for further studies. To be included in the meta-analysis, a study had to meet the following criteria: (1) The subject of the study must include plant species or assemblages; (2) the experimental treatment must consist of human trampling; (3) the study outcomes must include relative vegetation cover (RVC), or allow this metric to be derived; (4) the study must include controls and replication in randomized experimental designs linking cause and effect, with high strength of inference to non-study situations (*Cole & Bayfield, 1993*; *Sun & Walsh, 1998*). Studies which only met criteria (1) and (2) were also retrieved and summarised (Supplemental Information 3 and 4). Articles were assessed for relevance by one reviewer reading article titles and abstracts; a random subset of 40 from 304 identified articles was independently assessed for relevance by a second reviewer. Cohen's Kappa test showed the selection of relevant articles to have high inter-reviewer agreement ($K = 0.818$).

### Data extraction and analysis

One reviewer extracted data from graphs or tables; where data were not presented, or could not be derived, attempts were made to contact authors for the original data. Data retrieved from authors within twelve months of the enquiries being made were included in the meta-analysis. In one instance, data were converted from absolute cover to RVC (*Ikeda, 2003*). Due to the presentation of vegetation cover data as RVC, certain aspects of the effect of trampling on vegetation could not be directly investigated. Effect sizes used in meta-analysis usually summarise the effect of a treatment or intervention by comparing treatment and control means and variability before and after the treatment in question (*Egger, Smith & Altman, 2001*). However, data presented as RVC are relative to mean initial vegetation cover: measures of initial variability prior to trampling, necessary for effect size calculation, are not generally presented by the authors of primary studies. This means that the immediate post-trampling impact on vegetation (a typical measure of resistance) cannot be used as the dependent variable in meta-analyses, because appropriate data summarising the pre-trampling state of the vegetation cannot be extracted from the RVC metric. Therefore, we investigated differences in RVC between the period immediately after trampling and the time-point farthest from the trampling application recorded in each study, thus increasing the predictive power and increasing independence where there was a choice of time ranges in a study. Because primary studies often investigated more than one habitat, trampled at more than one intensity, single effect size estimates (hereafter referred to as 'trials') were extracted for any given habitat and trampling intensity pair within a single study. Investigating the presence of a recovery period, irrespective of length, as the main effect also increased the independence of trials within studies, giving each trial its

own comparator rather than comparing several trampling intensities to a single control. Thus, as other reviewers have found practical (*Yorks et al., 1997*), we investigated vegetation recovery as the dependent variable on a per trial basis, and defined this as the resilience of the vegetation. It should be noted that the terms resilience and resistance have been used elsewhere to refer to indices that combine data on both vegetation response and trampling intensity to produce a metric estimating what has been defined as the 'vulnerability' of any given site or vegetation type (*Cole & Bayfield, 1993*).

The main effect of the presence of a recovery period on trampled vegetation, calculated as the mean RVC at the final monitoring point of all replicates within a trial, minus the mean RVC immediately after trampling for the replicates, and other reasons for variability in vegetation recovery (trampling intensity, resistance, length of recovery period, Raunkiaer life-form, and habitat) were explored using meta-analysis and meta-regression (*Deeks, Altman & Bradburn, 2001*; *Gurevitch & Hedges, 2001*). Cohen's *d* effect sizes (*Deeks, Altman & Bradburn, 2001*) representing the change in vegetation cover between initial post-trampling monitoring and the final monitoring time-point were derived from the two RVC means with standard deviations and sample sizes (where the sample size is the number of experimental replicates, not the number of sub-sampled plots within replicates). Cohen's *d* uses a pooled estimate of standard deviation, allowing for the paired nature of the data points. Data were pooled and combined across trials using DerSimonian & Laird random effects meta-analysis based on standardised mean difference (SMD; *DerSimonian & Laird, 1986*; *Cooper & Hedges, 1994*). Meta-analysis combines the main effects from individual trials into a single estimate, whilst also taking the precision of each estimate into account (*Gurevitch & Hedges, 2001*). Meta-analysis also increases statistical power and allows the quantification and, where possible, exploration of variation between trials (*Deeks, Altman & Bradburn, 2001*; *Gurevitch & Hedges, 2001*). The random effects model assumes that there is variation amongst the true trial effects, and the aim of the analysis is to quantify such variation in the effect parameters; it is therefore appropriate for ecological questions where the true effect is likely to vary between trials (*Gurevitch & Hedges, 2001*; *Stewart, 2010*).

The effect of a post-trampling recovery period on plant growth as quantified by changes in RVC (i.e., vegetation resilience) was examined via the visual inspection of the forest plot of the estimated main effects from the trials, along with their 95% confidence intervals, and by formal tests of heterogeneity undertaken prior to meta-analysis (*Thompson & Sharp, 1999*). Publication bias was investigated by examination of funnel plot asymmetry and the Egger test (*Egger et al., 1997*). The relationships between the effect of a recovery period and the explanatory variables (length of recovery period, resistance and trampling intensity) were tested using random effects SMD meta-regressions in Stata v. 8.2 (*Stata Corporation, 2003*) using the program Metareg (*Sharp, 1998*). Meta-regression investigates the explanatory power of covariates for the observed pattern of main effect sizes; the random effects model acknowledges the potential for residual heterogeneity not explained by the covariate(s) between trials, therefore corresponding to random effects meta-analysis in assuming that the between-trial variance is not zero (*Thompson & Higgins, 2002*).

Meta-regressions were limited to continuous trial-level variables taking a range of values, and, to avoid data-dredging, were specified *a priori* (*Thompson & Higgins, 2002*) as resistance, length of the recovery period, and trampling intensity. Resistance was coded for meta-regressions as the mean RVC across replicates for each trial immediately after trampling had been applied. Collinearity between independent variables was investigated prior to performing the multiple meta-regressions (*Zuur, Ieno & Elphick, 2010*).

The use of subgroup analyses allows the investigation of variation in the main effect amongst particular groups of trials that are hypothesised to be biologically significant (*Brookes, Whitley & Peters, 2001*). We used subgroup analyses to explore variation in the effect of a recovery period amongst plant communities of different Raunkiaer life-forms and different habitats. Raunkiaer life-form subgroups, delineated by the position of the perennating buds (*Kent, 2012*), were: (juvenile) phanerophytes; chamaephytes; hemicryptophytes; geophytes; helophytes (where vulnerable to trampling); and therophytes. The life-form category of the plant species with the highest mean percentage cover, i.e., the community dominant, was taken as the category for a trial. Broad habitat categories were: alpine or tundra; temperate coniferous forest; temperate deciduous forest; subalpine and montane grass or shrubland; temperate shrubland; and temperate grassland. *Post hoc* within-subgroup multiple meta-regressions were also used to investigate the differential explanatory power of our *a priori* potential effect modifiers (resistance, length of the recovery period, and trampling intensity) for each life-form; collinearity between independent variables was also checked for all within-subgroup multiple meta-regressions (*Zuur, Ieno & Elphick, 2010*).

To investigate the potential effects of non-independence between trials on our results, due to, for example, individual studies conducting multiple trials testing different trampling intensities in similar habitat types, we also performed sensitivity analyses for all meta-regressions using robust variance estimation (*Hedges, Tipton & Johnson, 2010*). Robust variance estimation was performed according to *Hedges, Tipton & Johnson (2010)* on DerSimonian & Laird random effects meta-regressions conducted using the metafor package (*Viechtbauer, 2010*) for the statistical software R v. 3.0.2 (*R Core Team, 2005*).

## RESULTS

### Systematic search results

Searching and retrieval were conducted between October 2005 and July 2006. Three hundred and four articles were judged relevant at the title level; 145 articles remained in the systematic review after the title and abstract filter stage. Of the 145 only 24 were randomised controlled experiments eligible for inclusion in the final meta-analysis. Of these 24, nine presented data that could be extracted, or which were provided by the authors (Table 1). These nine contributed 188 trials to the meta-analysis. A methodological overview of articles not included in the meta-analysis, but which used a comparator or control in their experimental design, is provided in Supplemental Information 3. Lists of studies not using a comparator or control, and of those studies which could not be

**Table 1** Characteristics of studies included in the meta-analysis.

| Study | Habitat(s) | Trampling intensities (passes) & disturbance type | Trial sample size | Final follow-up time (years) | No. of effect sizes contributed by Raunkiaer subgroup | |
|---|---|---|---|---|---|---|
| Cole & Bayfield (1993) | Temperate coniferous forest, subalpine grassland | 25, 75, 200, 500, 700 (Acute) | 4 | 1 | Chamaephyte | 4 |
| | | | | | Hemicryptophyte | 4 |
| | | | | | Geophyte | 4 |
| Cole (1995b) | Alpine, subalpine/montane grass/shrubland, temperate coniferous/ deciduous forest | 10, 25, 50, 75, 100, 200, 250, 500 (Acute) | 4 | 1 | Chamaephyte | 17 |
| | | | | | Hemicryptophyte | 40 |
| | | | | | Geophyte | 15 |
| Cole & Spildie (1998) | Temperate coniferous forest | 25, 150 (Acute) | 4 | 1 | Chamaephyte | 2 |
| | | | | | Geophyte | 2 |
| Monz et al. (2000) | Subalpine/montane grass/ shrubland | 25, 75, 200, 500, 800 (Acute) | 4 | 1 | Hemicryptophyte | 16 |
| Cole & Monz (2002) | Alpine, subalpine/montane grass/shrubland, temperate coniferous/ | 75, 225, 600, 1500, 2400, 3000 (Chronic over 3 years) | 4 | 3 | Chamaephyte | 8 |
| | | | | | Hemicryptophyte | 12 |
| Monz (2002) | Alpine/tundra | 25, 75, 200, 500 (Acute) | 4 | 4 | Chamaephyte | 4 |
| | | | | | Helophyte | 4 |
| Ikeda (2003) | Temperate grassland | 80, 160, 640, 2560 (Chronic over 3.3 years) | 3 | 3.3 | Therophyte | 4 |
| Gallet, Lemauviel & Rozé (2004) | Temperate shrubland | 100, 200, 400, 800 (Acute, and chronic over 0.2 years) | 3 | 0.2 | Phanerophyte | 32 |
| Roovers et al. (2004) | Temperate deciduous forest, temperate shrubland | 25, 75, 250, 500 (Acute) | 4 | 2 | Phanerophyte | 8 |
| | | | | | Chamaephyte | 12 |

retrieved within the resource constraints of the project, are also provided for the benefit of future reviewers (Supplemental Information 4).

## Meta-analyses

### Post-trampling vegetation recovery

The 95% confidence intervals (horizontal arms) of the 188 trial effect sizes estimating post-trampling recovery of vegetation cover (resilience), via changes in RVC, are shown in a forest plot (Fig. 1); the weighted central point estimates have been omitted for clarity. An enlargeable PDF version of Fig. 1 with trial identifiers, trampling intensities and Cohen's $d$ effect sizes (with 95% confidence intervals) is also provided for closer inspection (Supplemental Information 2). The broken vertical line shows a positive, significant change in RVC after a period of recovery across all trials (SMD $= 1.357$, $z = 11.13$, $p < 0.001$; Fig. 1). The range of variation in the contributing trials gave significant variation in effect size ($\chi^2 = 576.83$, d.f. $= 187$, $p < 0.001$), with trials located both sides

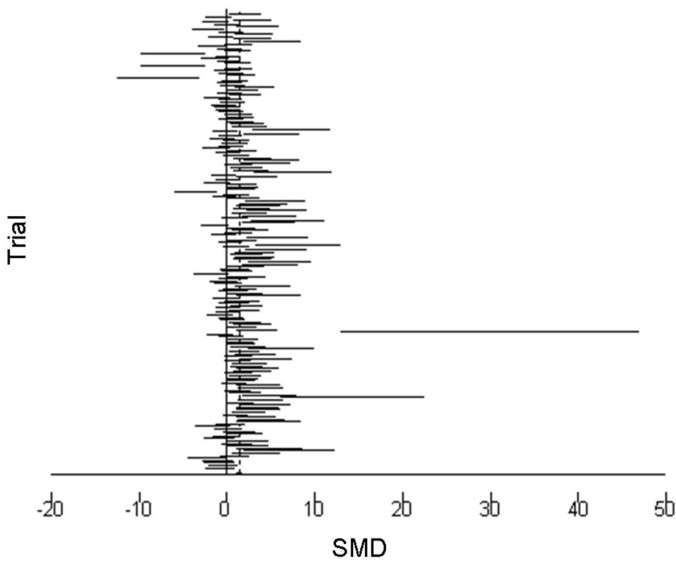

**Figure 1 Post-trampling recovery of vegetation.** A forest plot of the 188 effect size estimates pooled using random effects meta-analysis. Solid horizontal lines represent trial 95% confidence intervals. Trial confidence intervals not crossing the zero line of no effect indicate a positive or negative significant effect at the 5% level. Individual central point estimates are omitted for clarity.

of the line of no effect (no recovery), and with some trials exhibiting wide confidence intervals. Funnel plot asymmetry (Fig. 4) and the Egger test (*Egger et al., 1997*) suggest that there is potential bias in the data set, as there are fewer small negative trials than would be expected by chance (Egger bias $= 4.7$, $p < 0.001$).

*Multiple meta-regressions and subgroup analyses: effects of covariates*
The multiple meta-regression found that significant heterogeneity in vegetation recovery was explained by initial vegetation resistance (coeff. $= -0.357$, $z = -8.77$, $p < 0.001$), and the length of the recovery period (coeff. $= 0.246$, $z = 2.00$, $p < 0.045$); however, there was less evidence for an effect of trampling intensity (coeff. $= -0.001$, $z = -1.74$, $p = 0.082$). Correlations (Pearson's $r$) between independent variables were all below 0.35.

Subgroup analyses were conducted on Raunkiaer life-form categories based on the dominant field layer vegetation of each trial (Fig. 2). Vegetation dominated by phanerophytes ($n = 40$; SMD $= -0.033$, $z = 0.63$, $p = 0.528$), chamaephytes ($n = 47$; SMD $= 0.288$, $z = 1.15$, $p = 0.248$), helophytes ($n = 4$; SMD $= 1.918$, $z = 1.63$, $p = 0.102$) or therophytes ($n = 4$; SMD $= -0.737$, $z = 1.71$, $p = 0.087$), did not show a significant effect of the presence of a recovery period on RVC (95% CIs cross zero; Fig. 2); however, all of these life-form groupings displayed significant within-subgroup heterogeneity ($p < 0.01$). Hemicryptophytes ($n = 72$; SMD $= 1.955$, $z = 12.59$, $p < 0.001$) and geophytes ($n = 21$; SMD $= 1.660$, $z = 4.59$, $p < 0.001$) both showed a positive, significant main effect of a recovery period on RVC (95% CIs do not cross zero; Fig. 2), but also displayed significant heterogeneity ($p < 0.01$).

For chamaephytes, hemicryptophytes and geophytes, *post hoc* within-group meta-regressions examined the effects of resistance, length of a recovery period and trampling

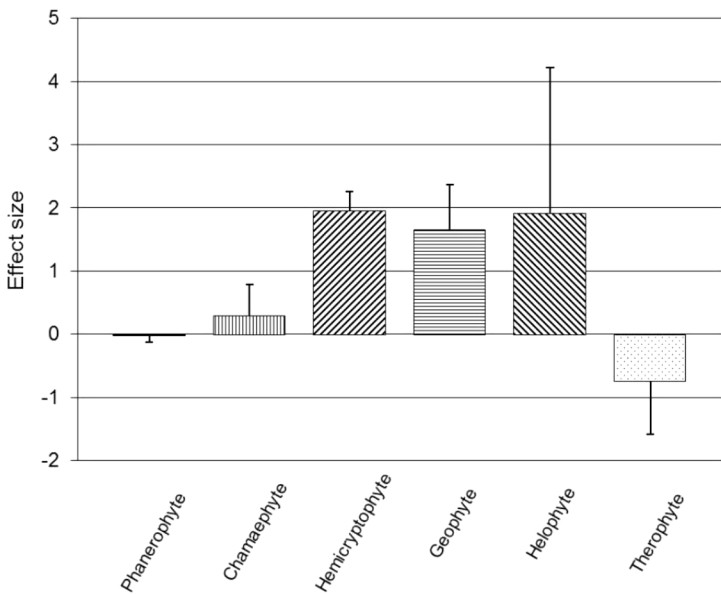

**Figure 2 Post-trampling recovery of vegetation for Raunkiaer subgroups.** Point estimates of effects sizes with 95% confidence intervals for Raunkiaer life-form subgroup analyses.

**Table 2 Outcomes of multiple meta-regressions for Raunkiaer life-form subgroup analyses.**

|  | Resistance | | | Recovery time | | | Trampling intensity | | |
|---|---|---|---|---|---|---|---|---|---|
|  | Coeff. | $z$ | $p$ | Coeff. | $z$ | $p$ | Coeff. | $z$ | $p$ |
| Chamaephytes | −0.300 | −3.69 | <0.001 | 0.495 | 1.93 | 0.054 | −0.001 | −1.16 | 0.247 |
| Hemicryptophytes | −0.233 | −4.74 | <0.001 | −0.374 | −1.95 | 0.051 | 0.001 | 0.97 | 0.330 |
| Geophytes | −0.411 | −3.41 | 0.001 | | N/A | | 0.002 | 1.26 | 0.206 |

intensity on vegetation recovery. Across the meta-regressions, correlations (Pearson's $r$) between independent variables were all below 0.52. The meta-regressions found the initial resistance of the vegetation to have a small but strong negative correlation with resilience for chamaephytes and hemicryptophytes (Table 2); the length of the recovery period was less important, but positive for chamaephytes and negative for hemicryptophytes (Table 2). Recovery time could not be included in the geophyte meta-regression due to all data points being reported one year after trampling; however, resistance was found to have a small, but significant, negative correlation for geophytes (Table 2). Trampling intensity was non-significant for chamaephytes, hemicryptophytes and geophytes (Table 2); the residual variation was significant for all three life-forms (chamaephytes: $p = 0.044$; hemicryptophytes: $p < 0.001$; geophytes: $p = 0.001$). Simple meta-regressions for trampling intensity within these subgroups also found no effect for hemicryptophytes ($p = 0.638$) and chamaephytes ($p = 0.883$), suggesting that the finding of no effect of trampling intensity was not being confounded by other covariates; however, trampling intensity had a significant effect on geophytes ($p = 0.001$), suggesting

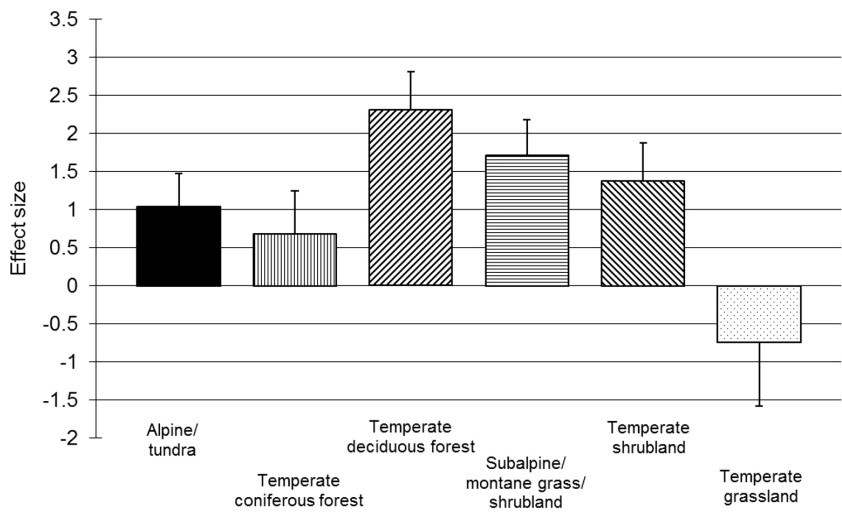

**Figure 3 Post-trampling recovery of vegetation within broad habitat types.** Point estimates of effects sizes with 95% confidence intervals for habitat subgroup analyses.

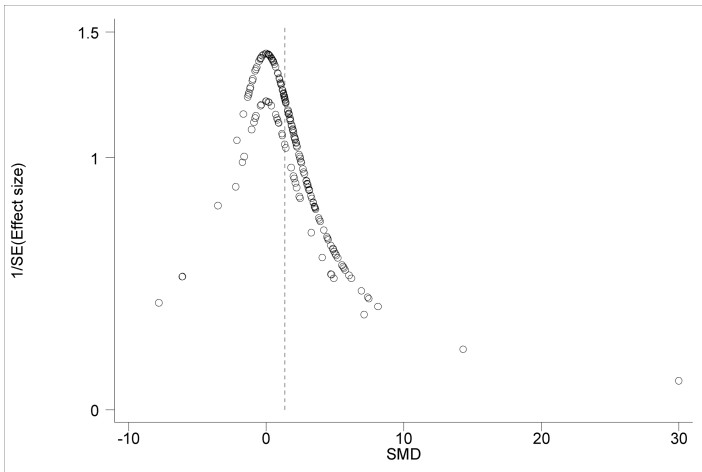

**Figure 4 Funnel plot of main effect size estimates.** A funnel plot of 188 main effect size estimates in relation to inverse variance. Funnel plot asymmetry illustrates that there are fewer small negative effect sizes than small positive effect sizes indicating a possible publication bias.

a stronger relationship (i.e., confounding variation) between trampling intensity and vegetation resistance in this subgroup. Habitat-based subgroup analyses revealed positive and significant main effects ($p < 0.05$; Fig. 3) and significant heterogeneity ($p < 0.01$) for all habitats, except temperate grassland, for which power was very low ($n = 4$).

### Sensitivity analyses

The use of robust variance estimation to account for the potential non-independence of trials within studies reduced the significance of all covariates tested. For the main multiple meta-regression, initial vegetation resistance (coeff. $= -0.039$; $t = -8.16$, $p < 0.001$) remained significant, whilst both length of the recovery period (coeff. $= 0.262$; $t = 1.80$,

$p = 0.13$) and trampling intensity (coeff. $= -0.001$; $t = -1.38$, $p = 0.226$) became non-significant. For the Raunkiaer subgroup analyses, hemicryptophytes were the only group for which a significant effect of the presence of a recovery period on RVC remained after adjustment for potential trial non-independence. The effects of robust variance estimation on the Raunkiaer life-form within-subgroup meta-regressions could only be investigated for chamaephytes; robust standard errors for the hemicryptohyte and geophyte subgroups could not be produced due to the small number of studies involved (4 and 3 respectively; *Hedges, Tipton & Johnson, 2010*). The effect of robust standard errors on the chamaephyte multiple meta-regression was to make all covariates non-significant: resistance: coeff. $= -0.033$; $t = -2.60$, $p = 0.122$; recovery time: coeff. $= 0.478$; $t = 1.18$, $p = 0.359$; trampling intensity: coeff. $= -0.001$; $t = -0.49$, $p = 0.672$.

## DISCUSSION

### Meta-analyses

We have found that, over the human trampling studies synthesized, vegetation generally recovered to some extent: there is an average significant positive effect of the presence of a recovery period on relative vegetation cover (RVC). However there is also significant heterogeneity, exceptions to this trend are frequent, and the effect may be over-estimated as a result of bias. Our results suggest that the initial resistance of a plant community, and the length of the recovery period, may be better predictors of vegetation resilience than the intensity of trampling undergone; that is, intrinsic properties of vegetation appear to be some of the most important determinants of resilience, with the magnitude of the actual disturbance explaining much less of the community response. The absence of a relationship between resilience and trampling intensity within the main meta-regression may be due to confounding variation caused by the differential vulnerability of different vegetation types, therefore these results are equivocal. However, the fact that *post hoc* life-form subgroup multiple meta-regressions for chamaephytes ($n = 47$) and hemicryptophytes ($n = 72$) also showed a lack of an effect of trampling intensity on resilience, supports the interpretation that intrinsic factors, i.e., plant traits, are often likely to be of primary importance for determining the vulnerability of vegetation to trampling (*Cole, 1995b*). The within-life-form subgroup meta-regressions confirmed that, except possibly for geophytes, life-form type was not obviously confounding the finding of no effect of trampling intensity on resilience within the main analysis. Overall, these results support a situation where particular plant functional traits are likely to be more important than projected intensity of use when considering the siting of recreational activities involving human trampling. This somewhat surprising result has important management ramifications because it suggests that even relatively low intensity trampling could be as damaging as high intensity trampling in certain plant communities. Thus, trampling may sometimes be unsustainable for vulnerable vegetation, potentially creating conflict between even relatively limited access and plant species- or community-focused conservation objectives.

The importance of initial vegetation resistance for recovery, indicated by the multiple meta-regressions, is likely to be a partial reflection of the negative correlation expected between resistance and resilience where vegetation is able to recover, and confirms the main effect in showing that recovery is the typical response for the levels of disturbance investigated in the studies included in our meta-analysis. However, the use of the relative metric RVC means that the recovery predicted for an impacted stand of vegetation is dependent on the initial absolute vegetation cover: a stand with high absolute cover and high resistance may not have much potential for recovery (restricted to 100%); this means that our analysis may slightly underestimate the typical recovery of more heavily impacted stands. However, it is arguably the relative resilience of different vegetation communities that is of greatest importance for informing sustainable management. In this respect our subgroup analyses confirmed the importance of Raunkiaer life-form (*Cole, 1995b*) across the 188 trials investigated, suggesting that hemicryptophytes and geophytes will be more resilient to trampling impacts relative to other life-forms. In contrast, chamaephyte-dominated vegetation did not show a main effect of recovery; indeed, chamaephyte-dominated communities have been shown to die-back after trampling disturbance, despite initially high resistance (*Cole, 1995b*; *Cole & Monz, 2002*). The negative relationship between resistance and resilience in the chamaephyte subgroup meta-regression reflects vegetation die-back after initially high resistance, rather than re-growth after initially low resistance. However, the chamaephyte subgroup meta-regression also showed a positive correlation of recovery time with resilience, suggesting that limited recovery may occur after die-back, given a period free from further disturbance. The negative correlation of recovery time for the hemicryptophyte subgroup may indicate that where recovery is not observed in the shorter-term, other factors, changes to soil characteristics for example, could make full recovery less likely.

The investigation of habitat as a reason for heterogeneity did not reveal any clear differences between subgroups (Fig. 3); however, the classification system used was broad, and sample sizes were small for some categories. As the vulnerability of vegetation to trampling may be related to primary productivity (*Liddle, 1975b*), a greater correlation between habitat and vegetation response might have been expected. The absence of a relationship in our results may be due to subtle biases relating to the distribution of trampling intensities and the length of recovery periods, confounding with life-form and the small number of replicates per habitat in our dataset. It is also possible that RVC does not reveal differences in production across habitat types as effectively as other measures, such as vegetation height or biomass.

## Critical analysis

We have found that the nature of the primary data available for meta-analytical synthesis of trampling impacts on vegetation requires a cautious approach to interpretation. Because of the lack of any measure of initial cover or variability in the majority of the primary studies analysed, we have focused on the effect of the presence of a period of recovery on post-trampling vegetation change, i.e., resilience. It should be remembered that because

RVC is calculated relative to initial vegetation cover, similar RVC changes may not equate to equivalent absolute changes in vegetation cover. Additionally, a non-significant main effect may either represent a lack of vegetation recovery, or a plant community with high resistance and therefore little potential for recovery; this is also a problem with using RVC to measure resilience on a study-level basis (*Cole & Bayfield, 1993*). This problem could be ameliorated by ensuring analyses of resilience refer back to primary data, possibly by investigating the absolute resistance of vegetation as an explanation for heterogeneity in recovery. The availability of raw data in Supplemental Information or data repositories, or higher response rates from authors of primary studies, would increase the value of trampling studies for synthetic, predictive research in conservation, and for research on the relationship between plant traits and disturbance. The availability of raw data would also allow a more powerful analysis using a hierarchical modeling approach, with increased power to detect real differences between groups, improved ability to explore interaction within and between trials, and would also allow for formal explorations of model choice (*Stewart, Altman & Askie, 2012*). Hierarchical modelling is also an alternative to robust variance estimation in accounting for the potential non-independence of trials within studies.

Our analysis is also limited by the characteristics of the primary studies. Most of the studies investigated several trampling intensities within single habitats, and some habitats and species were over-represented across studies. We therefore had to balance the competing biases of aggregation and non-independence in our synthesis. Non-independence can cause the overestimation of significance levels and the underestimation of confidence intervals (*Gurevitch & Hedges, 2001*); however, pooling effects across dependent and independent variables allows the investigation of interesting ecological heterogeneity, and increases the strength of inference to non-study situations, essentially in the same way as generalizing over subjects within studies (*Rosenthal, 1991*). The sensitivity analysis undertaken here suggests that, if non-independence is a strong property of the data we have analysed, then the most robust conclusions are those for the effects of recovery time and initial vegetation resistance in the main meta-regression.

The most important shortcoming exposed by our systematic review is possibly the lack of high-quality experimental information. Clearly, as we have found, restricting accepted information to randomized controlled experiments reduces the proportion of the literature which can be included in a meta-analytic review. This means that certain trends are confounded; for example, Raunkiaer subgroup analyses occasionally resulted in life-forms at certain time-points originating from a single study. Additionally, only one of the admitted studies examined long-term, chronic trampling, and then only over three years (*Cole & Monz, 2002*). This means that our conclusions are less certain for those situations in which chronic trampling impacts affect the physical, chemical and biological properties of the soil, subsequently affecting vegetation growth and succession over a longer period (*Burden & Randerson, 1972*). However, it seems highly likely that negative impacts are certain at high-levels of chronic trampling, and therefore potentially of less importance for evidence-based management seeking to balance biodiversity conservation

interest and lower-intensity access to open sites. This is a key question for societies in which open access to sites is becoming more common (*Sutherland et al., 2006*; *Marren, 2013*). The inclusion of experimental designs with lower internal validity does not necessarily increase external validity, and can simply result in more uncertainty (*Stewart, 2010*). This is a compelling reason for excluding relevant but low quality data from statistical analyses, and reflects the conclusions of other workers who have reviewed trampling impacts (*Yorks et al., 1997*). However, meta-analytical techniques designed to handle variable quality data are under development, and could prove useful for synthesizing such data in the future, provided that the uncertainty associated with lower-quality methodologies is adequately expressed. Bayesian Belief Networks have been utilised in ecology in such situations (*Newton et al., 2007*).

In attempting this synthesis we have observed that the research methodologies and reporting of existing experimental trampling studies may be inadequate for the underpinning of scientific management of plant species and communities of conservation concern under potential threat from recreational access. Changes in species composition or richness are not routinely reported, meaning that decision-making may be based on vegetation cover indicators, such as RVC, that may not approximate to full ecological recovery to pre-disturbance conditions (*Hylgaard, 1980*). Furthermore, the measure of RVC may be misleading as it lacks information on initial absolute plant cover, which may be of increased importance for the consideration of chronic trampling impacts. Whilst the standardisation of methods reduces unexplained variation, and increases comparability between studies, presenting results in a way which precludes the efficient meta-analysis of important responses is counter-productive to the aims of synthesis; this may be especially true where such synthesis is important for disseminating ecological results to conservation practitioners managing sites with under-studied, or unstudied, vegetation communities. Researchers could address these issues by making raw data available through online Supplemental Information where they do not present it in manuscripts, and by increased consideration of the potential uses of their results for meta-analytical studies.

## Management implications

The evidence presented here, systematically accumulated across high-quality experimental studies, suggests that vulnerable vegetation of conservation value should not be trampled, irrespective of the projected intensity of use. The range of trampling intensities investigated in the primary studies synthesized here suggests that even moderate disturbance can have significant effects on plant communities. Simple indicators such as life-form of the community dominant may then be useful for rapid assessments of a community's vulnerability to recreational pressure. Zonation of recreation into high and low intensity usage, with 'honey-pots' located away from vulnerable vegetation, may be a more effective conservation strategy than encouraging moderately intensive but more widespread recreational usage; especially given that occasional use results in the development of informal path-networks which may subsequently encourage further disturbance (*Roovers et al., 2004*).

Sites of conservation importance dominated by phanerophytes, chamaephytes, helophytes or therophytes should not experience regular trampling disturbance if deleterious impacts are to be avoided. Reducing trampling intensities may not be effective where adverse impacts are already occurring, although we did find a negative relationship between initial resistance and resilience for chamaephyte-dominated vegetation (i.e., high initial impacts may be followed by some recovery). Conversely, the current evidence base suggests that vegetation dominated by hemicryptophytes and geophytes, life-forms with more protection for their perennating buds (*Kent, 2012*), recovers to a greater extent than vegetation dominated by other life-forms, and could therefore potentially be trampled more intensively, provided monitoring is undertaken to provide early warning of deterioration or unsustainable use.

## Future work

Systematic reviews and meta-analyses should be periodically revisited in order to incorporate new data and to test new hypotheses or analytical techniques (*Pullin & Stewart, 2006*; *Stewart, 2010*). Since the research reported here was undertaken, several new studies investigating trampling impacts have been reported. In contrast to the work summarised here, *Bernhardt-Römermann et al. (2011)* found no evidence for an effect of Raunkiaer life-form in their pan-European experimental study of trampling disturbance; whilst other recent studies have found new evidence supporting the importance of this plant trait (*Andrés-Abellán et al., 2006*). *Bernhardt-Römermann et al. (2011)* did report evidence for the importance of plant rosette-type, the categories of which used by *Bernhardt-Römermann et al. (2011)* are classified as hemicryptophyte subtypes (*Kent, 2012*), supporting the importance of perennating bud position and protection for plant responses to trampling disturbance. Increased availability of full plant community data in primary studies could enable future reviewers to estimate the relative abundance of plant traits within vegetation, providing further insights into the importance of Raunkiaer life-form in contrast to other community-weighted plant traits (*Violle, Navas & Vile, 2007*).

Given that the large total number of trampling studies that our systematic review uncovered (Supplemental Information 3 and 4) is still almost certainly an underestimate of the evidence-base (due to our only following-up studies referenced within studies to a depth of one remove) we suggest that further investigation of ways to extract the maximum information from published studies will be the most efficient way of confirming when life-form, or other plant functional traits, are likely to be important indicators of vegetation responses to trampling. For example, a risk ratio metric of the relative proportions of different plant functional groups allowed *Newton, Stewart & Myers (2009)* to summarize a much larger proportion of the primary literature on north-west European heathland management than would have been possible if community composition data were demanded. The increasing availability of global plant trait data (*Kattge et al., 2011*), suggests that the concepts and information presented here could be extended to test new hypotheses about the relationship between trampling disturbance and plant vulnerability. We suggest that conservation planners, practitioners, and ecologists with

an interest in vegetation trampling, should develop a global prospective collaboration to ascertain priority questions and establish standards for monitoring and data reporting. These activities would facilitate future synthesis and maximize the potential for scientific evidence to inform policy in the increasingly important area of research into human impacts on ecosystems.

## ACKNOWLEDGEMENTS

We would like to thank the authors of primary studies who responded to our enquiries; Jan Bengtsson and Andrew Pullin for constructive criticism on an earlier version of this manuscript and helpful suggestions for taking the work forward.

### Funding

This work was performed without funding.

### Competing Interests

Gavin Stewart is an Academic Editor for PeerJ; he is a keen fellrunner and therefore has an active interest in issues pertaining to access and human vegetation trampling impacts. Oliver Pescott is currently an employee of CEH Wallingford.

### Author Contributions

- Oliver L. Pescott and Gavin B. Stewart conceived and designed the experiments, performed the experiments, analyzed the data, contributed reagents/materials/analysis tools, wrote the paper, prepared figures and/or tables, reviewed drafts of the paper.

### Supplemental Information

Supplemental information for this article can be found online at http://dx.doi.org/10.7717/peerj.360.

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
