# Peer review of "Assessing the impact of human trampling on vegetation: a systematic review and meta-analysis of experimental evidence"

_PeerJ, doi:10.7717/peerj.360_

## Round 0.1 · original submission · Minor Revisions

Consider the suggestions by reviewer 1 and give arguments in case you do not find this suggestion valuable.

·

Basic reporting

The reporting of the paper is very clear and thorough both in terms of background and methods.

Experimental design

The research question is clear as are the methods used for conducting the meta-analyses and the meta-regression analyses. The methods used are of a high technical standard.

Validity of the findings

I think the conclusions are sound and based on high quality statistical analyses and well established systematic review methods.

I have one suggestion for improving the analyses but its relatively minor and up to the authors if they want to consider it:

I think it makes sense that you extracted 188 effect sizes from the 9 randomised controlled experiments - as this is a good use of the data reported in the papers. The discussion mentions the limitations of non-independence in the synthesis but I wonder whether its worth doing a sensitivity analyses using robust standard errors as a further way of justifying the validity of this approach. See for example: http://onlinelibrary.wiley.com/doi/10.1002/jrsm.5/pdf

·

Basic reporting

Clearly and concisely written throughout, this article conforms to the PeerJ structure whilst incorporating standard sections used in reporting the findings of systematic reviews and meta-analyses. The introduction and discussion are adequate to place the study aims and findings within the context of existing research and demonstrates how the study fills a knowledge gap in the subject. Figures are of sufficient quality for publication with the exception that the formatting of the final column in table 1. The column needs to be widened to properly accomodate the term 'hemicryptophyte'.

Experimental design

The investigation has been undertaken rigorously and meets the standard requirements of systematic review and meta-analysis.

Validity of the findings

As with all properly undertaken systematic reviews, the findings of this article not only attempt to summarise a general outcome relating to a given intervention but also shine a light on the needs of future research. The article is carefully argued throughout and therefore does not overstate the findings and is thorough in the treatment of the limitations of existing research. I hope the plant conservation community are alerted to the existence of this article and take its recommendations on board in order to improve future practice and research.

---

## Round 0.2 · accepted · Accept

Your manuscript can be accepted by PeerJ.